# Shifts in Diatom Dominance Associated with Seasonal Changes in an Estuarine-Mangrove Phytoplankton Community

**Fareha Hilaluddin [1], Fatimah Md. Yusoff [2,3],\* and Tatsuki Toda [4]**

[1]  Laboratory of Marine Biotechnology, Institute of Bioscience, Universiti Putra Malaysia (UPM), Serdang, Selangor 43400, Malaysia; hfareha01@gmail.com

[2]  International Institute of Aquaculture and Aquatic Sciences, Universiti Putra Malaysia (UPM), Port Dickson, Negeri Sembilan 71050, Malaysia

[3]  Department of Aquaculture, Faculty of Agriculture, Universiti Putra Malaysia (UPM), Serdang, Selangor 43400, Malaysia

[4]  Laboratory of Restoration Ecology, Faculty of Engineering, Soka University, 1-236 Tangi-cho, Hachioji, Tokyo 192-8577, Japan; toda@soka.ac.jp

\*  Correspondence: fatimamy@upm.edu.my

**Abstract:** A study on seasonal phytoplankton abundance and composition in a mangrove estuary, Matang Mangrove Forest Reserve (MMFR), Malaysia, was carried out to determine the phytoplankton structure in this ecosystem, and to identify potential indicators of environmental changes. Phytoplankton samples were collected bimonthly from June 2010 to April 2011, to cover both dry (June to October) and wet (November to April) seasons, at four selected sampling sites along the river. Diatoms showed the highest number of species (50 species) from a total of 85 phytoplankton species from 76 genera. Diatoms contributed more than 90% of the total phytoplankton abundance during the dry season (southwest monsoon) and less than 70% during the wet season (northeast monsoon) as dinoflagellates became more abundant during the rainy season. Two diatoms were recorded as dominant species throughout the sampling period; *Cyclotella* sp. and *Skeletonema costatum*. *Cyclotella* sp. formed the most abundant species (62% of total phytoplankton) during the dry period characterized by low nutrients and relatively low turbidity. *Skeletonema costatum* contributed 93% of the total phytoplankton in October, which marked the end of the dry season and the beginning of the wet season, characterized by strong winds and high waves leading to the upwelling of the water column. Massive blooms of *Skeletonema costatum* occurred during the upwelling when total nitrogen (TN) and total phosphorus (TP) concentrations were highest ($p < 0.05$) throughout the year. The abundance of diatom species during the wet season was more evenly distributed, with most diatom species contributing less than 12% of the total phytoplankton. Autotrophic producers such as diatoms were limited by high turbidity during the northeast monsoon when the rainfall was high. During the wet season, *Cyclotella* and *Skeletonema costatum* only contributed 9% and 5% of the total phytoplankton, respectively, as dinoflagellates had more competitive advantage in turbid waters. This study illustrates that some diatom species such as *Cyclotella* sp. and *Skeletonema costatum* could be used as indicators of the environmental changes in marine waters.

**Keywords:** diatoms; phytoplankton; mangrove estuary; monsoon seasons; nutrient out-welling; indicator species

## 1. Introduction

The mangrove ecosystem is known as a dynamic and highly productive area with rich diversity. This unique ecosystem is generally characterized by different mangrove species with different adaptive root systems that enable them to survive in harsh conditions with silty and endless ebbs of flowing water [1]. The ecosystem plays multiple ecological functions as a natural buffer that distillate and recycle nutrients, including protecting coastlines against natural disasters (e.g., storms, tsunamis, and wave actions) and also providing nursery grounds for fishes, birds, and other fauna [2,3]. Despite various benefits to flora and fauna, mangroves also provide significant contributions to humans with regard to economic and cultural aspects [4,5]. The increased nutrient inputs from agriculture activities, agriculture-based industries, and other land development pursuits eventually end up in the coastal waters. As contended by Nicholls et al. [6], mangroves grow faster in areas with higher nutrient inputs compared to areas with fewer nutrient sources. Some have suggested that increased mangrove cover can also occur due to direct anthropogenic impacts from salt pond construction and river damming [7]. However, high nutrient concentrations that could enhance the mangrove growth rate could actually increase the vulnerability of marine waters to environmental stresses due to eutrophication [8]. The dynamics of coastal zones indirectly involve climate changes, which may disrupt the mangrove system through higher sea-level rise, seawater acidification, storms, and also cause typical coastal problems such as flooding and erosion, as well as economic losses and human fatalities [9].

Mangrove estuaries are the freshwater links between the rivers and the seas, and they are subjected to strong tidal currents, changing water depth, varying salinity levels, and increasing sediment concentrations [10]. The mixing by tidal and wind causes water circulation, uplifting of nutrients from the bottom, and salinity gradients that form the main drivers influencing phytoplankton dynamics in an estuary. Together with favorable light conditions, an increasing amount of nutrients enhances phytoplankton growth. Tidal mixing and waves also bring dissolved oxygen to the bottom and oxidize the hypoxic layer [11]. In addition, the salinity gradient resulting from the tidal mixing contributes to the high species diversity in the estuary, as most species have developed tolerance across the entire salinity range. Roubeix et al. [12] reported that some marine species may disappear when salinity decreases, while some holo-euryhaline species such as the diatom *Cyclotella meneghiniana* are able to grow in increasing salinity levels. In addition, the energy subsidy from the surrounding mangrove forests, in the form of organic matter, is also critical for supporting various biota such as microorganisms, cockles, and mussels [13].

Phytoplankton indicators have been used as one of the significant metrics in identifying the health status of the mangrove ecosystem [14]. Some phytoplankton species are suitable indicators for the assessment of water quality as they respond quickly to changes in nutrient concentrations, salinity, pH, and other environmental factors related to water degradation [15,16]. The community structure of phytoplankton, including species composition, biodiversity, evenness, richness, and similarity indices have been used in biological monitoring, based on the assumption that a pristine ecosystem is more diverse, stable and resilient compared to degraded environments [17,18]. As a primary producer, phytoplankton plays a vital role in supporting various trophic levels in the aquatic food web [19]. According to San Martin et al. [20], the energy transfer efficiency of phytoplankton through various trophic levels was dependent on the productive gradient, which was characterized by the plankton community composition. This statement was supported by Armengol et al. [21], who reported that higher grazing rate was primarily confined to oligotrophic areas, probably due to higher phytoplankton diversity compared to productive areas characterized by lower species diversity with frequent algal blooms. Phytoplankton blooms have become a major global concern since some of the species associated with eutrophication can produce toxins that cause harmful effects to other organisms along the food chain, including livestock and humans [22,23]. Furthermore, phytoplankton bloom magnitude, frequency, and spatial extent have been used as significant metrics to indicate nutrient fluctuations from freshwater inflows into coastal areas [24].

As a major group of phytoplankton in the coastal ecosystem, diatoms are valuable indicators of the aquatic environment since they reproduce and respond rapidly to environmental changes and provide an early warning for pollution [25,26]. Diatoms react rapidly to environmental changes by shifting their community composition due to eutrophic conditions [27]. They also increase significantly in abundance along the gradient of increasing carbon dioxide [28] and other inorganic nutrients, especially nitrogen and phosphorus, the rates of which are dependent on specific diatom species [29]. Diatoms are known to dominate marine and estuarine waters, including mudflats [30–32]. Stickley et al. [33] suggested that diatoms are useful indicators in estuarine waters since they have specific adaptation and are able to provide clues to climate history. Periphytic diatoms such as *Pinnularia*, *Eunotia*, *Navicula*, *Gomphonema*, and *Nitzschia* are also associated with oligotrophic environments with specific ecological tolerances to pH, conductivity, and trophic preferences [34]. Kafouris et al. [35] showed that the efficiency of benthic diatoms as bioindicators of nutrient enrichment in oligotrophic coastal systems was related to the increased biomass of *Cocconeis* species. Some diatom species, such as *Licmophora,* could predominate epilithic communities in an oligotrophic bay, depending on the current pattern, temperature, oxygen, silicate, and salinity as the most important factors [36,37]. In fact, Shen et al. [38] reported that different regional current patterns determined the distribution of biogenic silica contents in the surface water, which in turn affected the primary production in the Philippines' coastal waters.

Although information on phytoplankton abundance and diversity in coastal areas has been reported, the potential use of diatom species as bioindicators in tropical coastal ecosystems is still limited. The present study aimed to illustrate the relationship between diatom species dominance with environmental factors that regulate the phytoplankton composition in an estuarine-mangrove ecosystem. This information is essential in understanding the succession of the diatom community that links to environmental changes due to eutrophication or seasonal/climate change. The examination of the species dominance in the estuarine-mangrove phytoplankton community would reveal the potential use of diatoms as bioindicators associated with environmental variations related to seasonal changes.

## 2. Materials and Methods

### 2.1. Study Areas

This study was carried out bimonthly from June 2010 to April 2011 in Sangga Kechil mangrove-estuary, from the northern part of the Matang Mangrove Forest Reserve (MMFR), Perak located at the western coast of peninsular Malaysia (4°50′ N, 100°35′ E) (Figure 1). The MMFR is the largest tract of a well-managed mangrove area in Malaysia, with a total area of 40,466 ha, and is dominated by silvicultured *Rhizophora apiculata* [39]. Sangga Kechil is a branch tributary of the Sangga Besar river (Figure 1), which is a major waterway for fishing boats and supports cage aquaculture and cockle cultivation [40], as the hydrology is dominated by semi-diel tidal circulation [41]. During the sampling periods, tidal levels were in the range of 0.1 m to 2.5 m (Figure 2). The current speed may exceed 1 knot throughout the year, and it is in the northeast direction through the central part of the Strait, resulting in lower tidal energy recorded in the northern part of the straits near MMFR compared to the southern part [42].

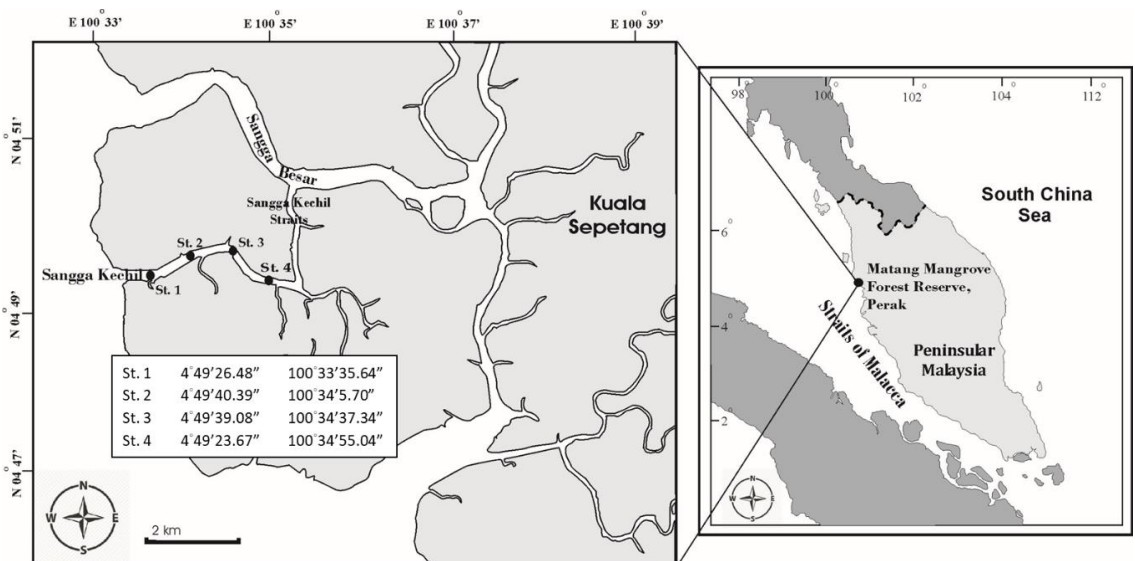

**Figure 1.** Map showing the locations of four selected sampling sites of the study area in the Sangga Kechil estuarine-mangrove area, Matang, Perak.

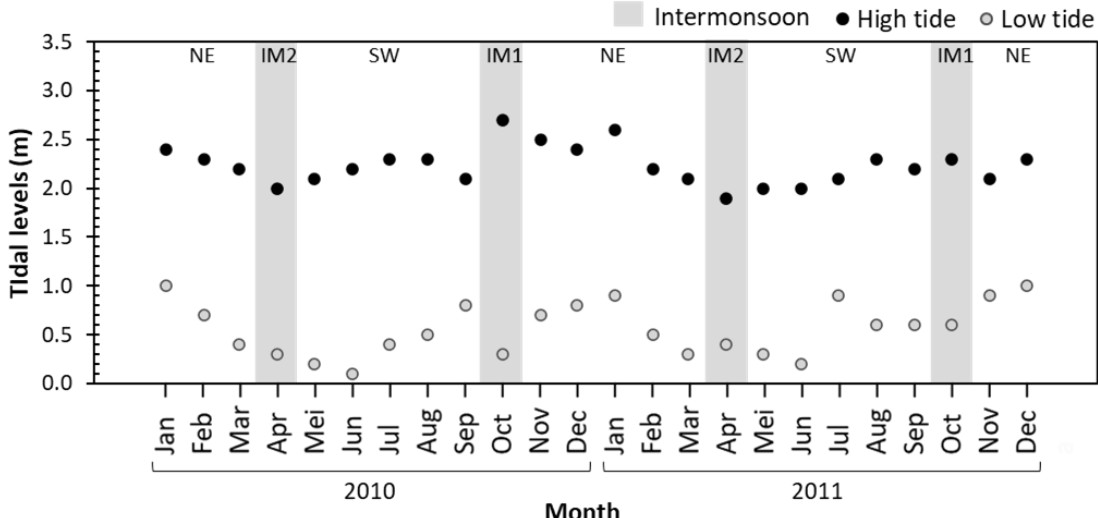

**Figure 2.** Tidal levels (m) at Sangga Kechil estuarine-mangrove from 2010 to 2011. NE, northeast; SW, southwest; IM1, intermonsoon (transition from dry to wet season); IM2, intermonsoon (transition from wet to dry season).

### 2.2. Samples and Environmental Data Collection

Water transparency was determined by a Secchi disk [43,44], while environmental parameters such as temperature, salinity, pH, and turbidity were measured in situ using a YSI Model 556 MPS handheld multiparameter (YSI Incorporated, USA). Water samples for chlorophyll *a* and nutrient analyses (TN, TP, soluble reactive phosphorus (SRP), total ammonium nitrogen (TAN), and nitrate+nitrite ($NO_3^- + NO_2^-$-N) were collected using a 5-L Niskin water sampler. Samples collected were transferred into wide-mouth polyethylene bottles and were kept in $-4\,°C$ ice chest upon returning to the laboratory. However, chlorophyll *a* samples were filtered immediately in the field at low vacuum pressure using Whatman GF/F glass fiber filters. A few drops of $MgCO_3$ were added during the filtration process, and each filter was folded and kept in aluminum foil bags in a $-4\,°C$ ice chest for transportation to the laboratory. The nutrient samples were further processed and analyzed in the laboratory.

### 2.3. Chlorophyll a, Nutrients and Rainfall Data

Upon arrival at the laboratory, chlorophyll *a* and nutrient samples were stored at −20 °C and were processed and analyzed immediately within one week of collection, according to spectrophotometric methods using UV-1601 UV Visible Spectrophotometer (Shimadzu, Japan). Chlorophyll *a* was extracted and standardized in 90% of acetone based on Parsons et al. [45]. Unfiltered nutrient samples were processed to determine the TN and TP using peroxodisulfate oxidation [46], while filtered water samples through Whatman®cellulose acetate membrane filters were used to analyze TAN, $NO_3^- + NO_2^-$-N) and SRP. Rainfall data during the sampling periods were obtained from the Malaysian Meteorological Department (MetMalaysia) based on measurements recorded at a nearby town of Taiping, located approximately 10 km to the east of MMFR (Figure 3).

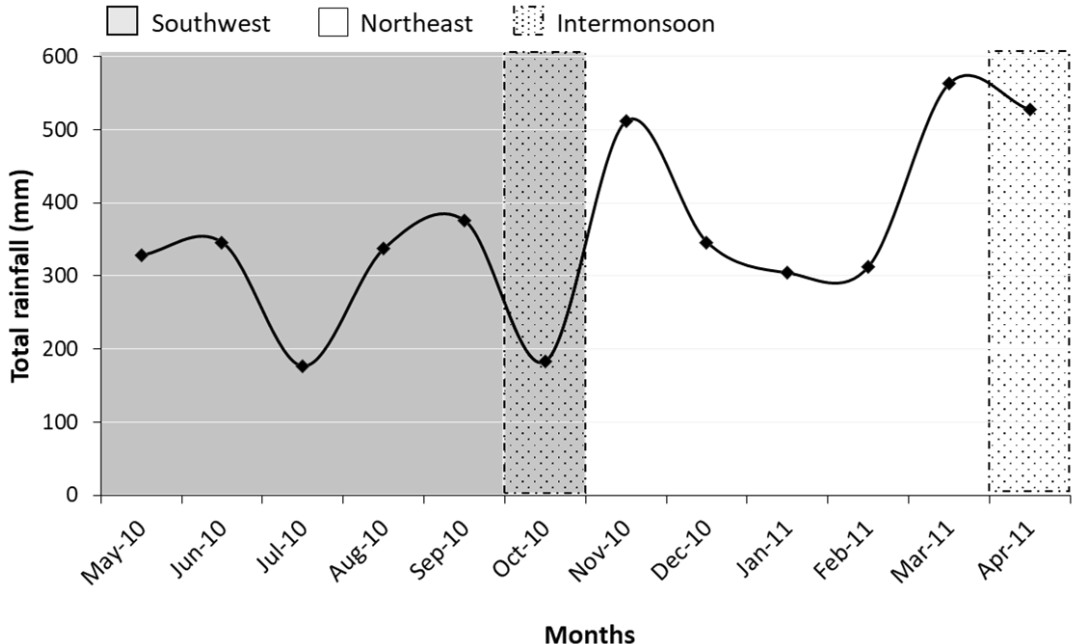

**Figure 3.** Rainfall data of the study area showing the dry season during the southwest monsoon (May to September) and wet season during the northeast monsoon (November to March) separated by inter-monsoons in October and April.

### 2.4. Phytoplankton Collection and Enumeration

Triplicate phytoplankton samples with 1 L each from the surface, middle, and bottom of the sampling station (based on of the water depth) were collected using a 5 L Niskin water sampler and were transferred into labeled polyethylene bottles. The samples were fixed at the sampling site using 6–8 mL acidic Lugol's iodine. In the laboratory, samples were allowed to settle for 24 h, and concentrated samples were transferred into 100 mL tight capped bottles and stored in the dark place to prevent light degradation before enumeration. All preserved samples were checked regularly and Lugol's iodine was added as required to ensure adequate preservative in each sample bottle. Phytoplankton enumeration in triplicate was completed using a Sedgewick-Rafter counting chamber under a compound microscope Nikon Eclipse E400 (Nikon, Japan) according to LeGresley and McDermott [47], and was counted in cell $L^{-1}$ based on Andersen and Throndsen [48]. For identification purposes, samples were also viewed under smart microscopy of ZEISS Axiolab 5 with a digital imaging process and references were made to the latest websites and books including *Diatoms of North America* (https://diatoms.org), AlgaeBase (http://www.algaebase.org), WORMS: World Register of Marine Species (http://www.marinespecies.org), and related phytoplankton taxonomic books [49,50].

*2.5. Data Analyses*

Spatial and seasonal physical/chemical patterns were illustrated using contour graphics generated using SURFER 13 (Surfer®, Golden Software, LLC). The relationship among various physicochemical and biological parameters were analyzed based on two-tailed Pearson correlation and canonical correspondence analysis (CCA). The interpretation of data on community structure and the relationships among samples through similarity measures were performed using PRIMER v7 (Plymouth routine in multivariate ecological research) [51]. Differences in biodiversity and abundance amongst seasons were analyzed using permutation multivariate analysis of variance (PERMANOVA) by comparing the actual *F* test result from random permutations. Additionally, pairwise comparisons among seasons were obtained by an additional run of the PERMANOVA routine using PRIMER v7. The most commonly used clustering techniques of hierarchical agglomerative methods of overall phytoplankton communities were done on square-root transformed data. Biodiversity indices were calculated using Shannon–Wiener diversity index (H′) and Pielou's evenness index (J′) to characterize the species diversity in a community across seasons.

## 3. Results

*3.1. Diatom Dominated Phytoplankton Population*

The phytoplankton community of Sangga Kechil estuarine-mangrove area consisted of 85 species from 76 phytoplankton genera, with 50 species of diatoms (Bacillariophyceae), 21 species of dinoflagellates (Pyrrophyceae), 10 species of green algae (Chlorophyceae), two species of cyanobacteria (Cyanophyceae) and one species each of silicoflagellates (Dictyochophyceae) and euglenoids (Euglenophyceae). The community was significantly ($p < 0.05$) more diverse (H′max = 4.16) during the southwest monsoon season with 64 species, compared to the northeast monsoon season with 54 species and inter-monsoonal periods with 53 and 34 species in October and April, respectively. The lowest number of species and the lowest diversity (mean H′ = 0.36) during inter-monsoonal periods in October coincided with the bloom of *Skeletonema costatum* (Figure 4). The highest diversity indices of H′ during the northeast monsoon (mean H′ = 3.04) showed a generally better distribution and balance of species abundance in a community, compared to the dry season with high species dominance (Figure 4).

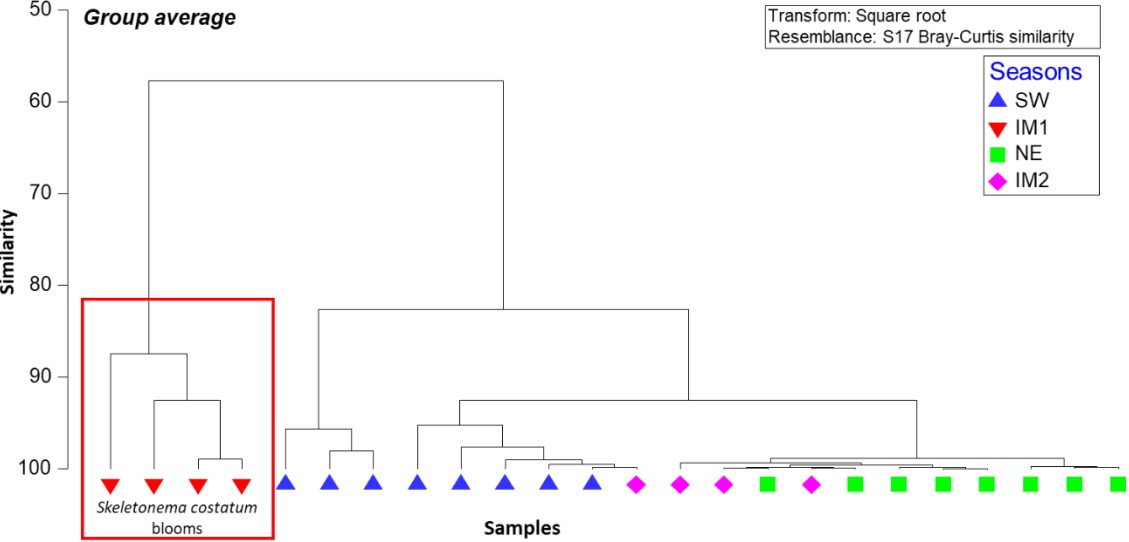

**Figure 4.** Hierarchical clustering of Shannon–wiener diversity index (H′) using group-average linking 24 phytoplankton samples of Bray Curtis dissimilarities according to seasons. A group of samples during the intermonsoon in October (IM1), separated at 60%, was associated with the massive blooms of *Skeletonema costatum*.

The hierarchical clustering amongst phytoplankton abundance across seasons at 50% dissimilarity showed four groups; each was clearly associated with a specific season (Figure 5). The clustering dendrogram displayed a strong inter-relation with high similarities within groups of similar species in specific seasons. Diatoms formed the highest percentage contributions compared to other phytoplankton groups with the occurrence of *Cyclotella* sp., and *Skeletonema costatum* dominating the southwest monsoon and the inter-monsoonal period IM1, respectively (Figure S1). Dinoflagellates became dominant during the late northeast season and the inter-monsoonal period in April (IM2), associated with relatively low temperatures and high total dissolved solids during the wet season (Figure 5 and Figure S1, and Table 1).

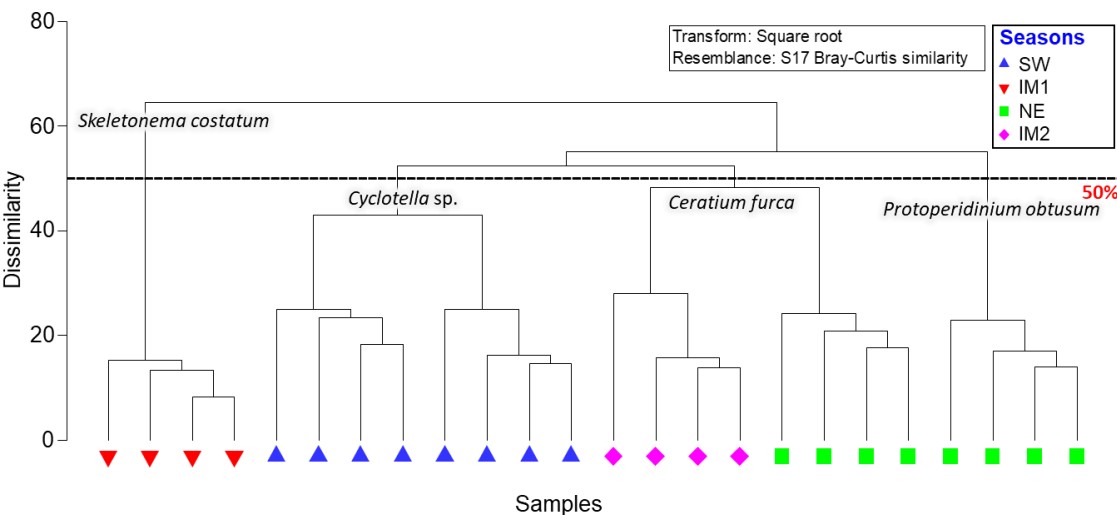

**Figure 5.** Hierarchical clustering of dendrogram showing dominant species in various clusters associated with seasons (group-average linking 24 phytoplankton samples of Bray Curtis dissimilarities calculated on square-root transformed densities). SW, southwest monsoon; NE, northeast monsoon; IM1, intermonsoon in October; IM2, intermonsoon in April.

Seasonal variations of the phytoplankton population (mean %) showed that the diatom was the most dominant group in all seasons except during the inter-monsoonal period in April (IM2) when dinoflagellates formed 56.13% of the total phytoplankton density (Figure 6). The mean cell densities of euglenoids, silicoflagellates, cyanobacteria, and green algae were relatively poor in this study with the absence of green algae during the inter-monsoonal period in October (IM1) due to massive blooms of a diatom species (Figure 6 and Figure S2). In fact, seasonal changes of diatoms showed a similar trend as the total phytoplankton densities, indicating that diatoms were the major force within the phytoplankton community in the area. However, during the inter-monsoonal period in April (IM2), the phytoplankton community was dominated by dinoflagellates groups, with a mean density of 41,866.7 cells $L^{-1}$ ± 17,545.8 compared to the diatoms with a mean density of 32,575.0 cells $L^{-1}$ ± 1504.1 (Figure S2). Among 50 diatom species, *Cyclotella* sp. was commonly found during the southwest season but was less abundant during the northeast monsoon (Figure 7). Another diatom, *Skeletonema costatum*, formed 93.48% of the total phytoplankton density during the inter-monsoonal period in October (IM1).

**Table 1.** Mean ± standard error (mean ± SE) and covariance correlation (Pearson) of major parameters in the Matang mangrove.

| Variables | Mean ± SE | 1. | 2. | 3. | 4. | 5. | 6. | 7. | 8. | 9. | 10. | 11. | 12. | 13. | 14. | 15. | 16. |
|---|---|---|---|---|---|---|---|---|---|---|---|---|---|---|---|---|---|
| 1. Temp. (°C) | 30.24 ± 0.21 | 1 | | | | | | | | | | | | | | | |
| 2. DO (mg L$^{-1}$) | 5.19 ± 0.13 | 0.48 | 1 | | | | | | | | | | | | | | |
| 3. Sal. (PSU) | 23.85 ± 0.40 | 0.14 | −0.00 | 1 | | | | | | | | | | | | | |
| 4. pH | 7.41 ± 0.07 | −0.41 | 0.02 | 0.34 | 1 | | | | | | | | | | | | |
| 5. TDS (g L$^{-1}$) | 24.61 ± 0.22 | −0.18 | −0.09 | **0.70** | 0.13 | 1 | | | | | | | | | | | |
| 6. Turb. (NTU) | 48.60 ± 12.12 | 0.39 | −0.27 | 0.10 | −0.30 | 0.04 | 1 | | | | | | | | | | |
| 7. TN (µg L$^{-1}$) | 144.73 ± 40.19 | 0.24 | −0.42 | 0.17 | −0.37 | 0.02 | **0.59** | 1 | | | | | | | | | |
| 8. TP (µg L$^{-1}$) | 159.68 ± 23.61 | 0.23 | −0.39 | 0.07 | −0.24 | −0.00 | **0.89** | **0.58** | 1 | | | | | | | | |
| 9. TAN (µg L$^{-1}$) | 18.45 ± 1.82 | 0.31 | 0.41 | −0.34 | −0.13 | −0.38 | −0.22 | **−0.59** | −0.32 | 1 | | | | | | | |
| 10. NO-N (µg L$^{-1}$) | 52.69 ± 14.03 | −0.41 | −0.12 | **−0.80** | −0.38 | −0.18 | −0.19 | −0.16 | −0.16 | 0.01 | 1 | | | | | | |
| 11. SRP (µg L$^{-1}$) | 42.46 ± 4.71 | −0.40 | −0.48 | −0.29 | −0.43 | −0.02 | −0.05 | 0.46 | 0.06 | **−0.51** | **0.54** | 1 | | | | | |
| 12. Phyto. (cell L$^{-1}$) | 288,813.89 ± 104,249.51 | −0.03 | −0.41 | 0.28 | −0.22 | 0.06 | 0.07 | **0.69** | 0.15 | **−0.64** | −0.15 | **0.69** | 1 | | | | |
| 13. Diatom (cell L$^{-1}$) | 270,986.11 ± 104,492.24 | −0.06 | −0.42 | 0.29 | −0.21 | 0.09 | 0.07 | **0.69** | 0.14 | **−0.66** | −0.15 | **0.69** | **0.99** | 1 | | | |
| 14. Dino. (cell L$^{-1}$) | 16,679.86 ± 3664.83 | **0.69** | 0.42 | −0.29 | −0.28 | **−0.59** | 0.22 | 0.10 | 0.15 | 0.35 | −0.16 | −0.25 | −0.06 | −0.10 | 1 | | |
| 15. Rainfall (mm) | 347.70 ± 23.32 | 0.40 | **0.63** | −0.37 | −0.02 | −0.43 | −0.31 | **−0.58** | −0.40 | **0.80** | 0.06 | **−0.62** | **−0.64** | **−0.65** | 0.44 | 1 | |
| 16. Tidal level (m) | 2.28 ± 0.05 | −0.46 | **−0.54** | 0.20 | 0.11 | −0.10 | 0.37 | 0.07 | **0.50** | 0.16 | **−0.77** | **0.83** | **0.72** | **0.74** | −0.44 | **−0.93** | 1 |

Values in bold indicate significant relationships at *p* < 0.05. Temp. = temperature, DO = dissolved oxygen, Sal. = salinity, Cond. = conductivity, TDS = total dissolved solids, Turb. = turbidity, TN = total nitrogen, TP = total phosphorus, TAN = total ammonium nitrogen, NO-N = nitrate ($NO_3^-$) + nitrite ($NO_2^-$), SRP = soluble reactive phosphorus, Phyto. = total phytoplankton, Diatom = total diatom, Dino. = total dinoflagellates.

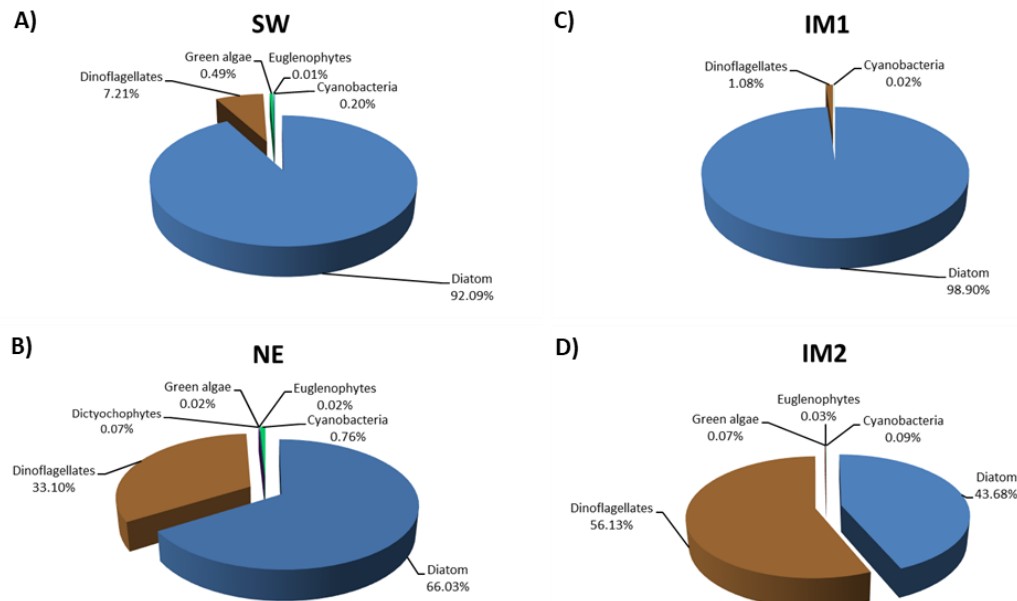

**Figure 6.** Seasonal contributions of different major phytoplankton groups (mean %) in the Sangga Kechil estuarine-mangrove area during (**A**) the dry season-SW: southwest monsoon, and (**B**) wet season–NE, northeast monsoon, and during the transitional intermonsoon periods (**C**) IM1 (October) and (**D**) IM2 (April).

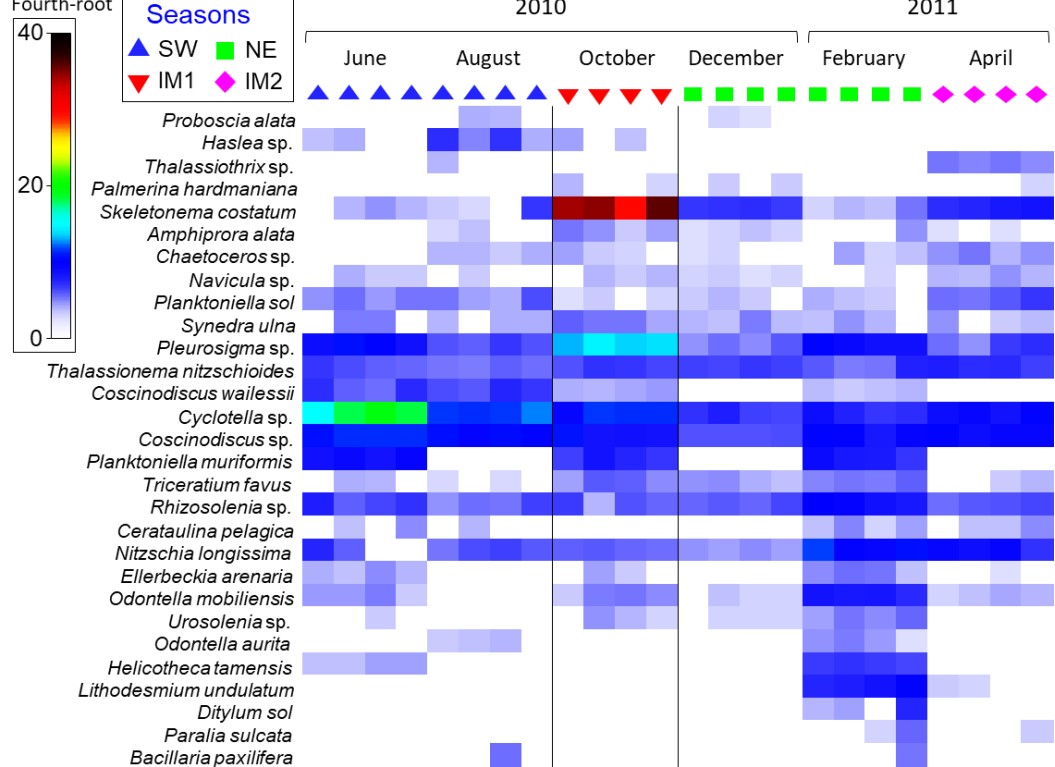

**Figure 7.** Shade plot, a visual representation abundances of a total of 50 diatom species out of 85 phytoplankton taxa across different stations and different monsoonal seasons with spectrum scale intensity proportional to the fourth-root abundance. The dendrogram shows species clustering using standard agglomerative methods of calculating group average, with white denotes the absence of the species.

### 3.2. Correlation of Physical and Chemical Parameters and CCA Ordination of Dominant Diatoms

The main driving force controlling changes of environmental factors in different seasons was the rainfall and tidal mixing that caused changes in various parameters. The heavy rainfall of $347.70 \pm 23.32$ mm (mean $\pm$ SE) was significantly correlated to the total phytoplankton ($-0.64$, $p < 0.05$) and diatoms ($-0.65$, $p < 0.05$) (Table 1) as the rainfall influenced the seasonal freshwater inflows into the coastal waters, and contributed to increased nutrients as well as causing changes to salinity, pH, turbidity, and other physical and chemical parameters in the estuary (Figures 8 and 9), which in turn affect the phytoplankton distribution in the area. Based on Table 1, correlations between nutrient loadings and total rainfall were highly significant ($p < 0.05$), especially with TAN (0.80 at $p < 0.05$). Diatoms were positively correlated with the TN and SRP ($p < 0.05$) (Table 1). In addition, heavy rainfall during the northeast monsoon caused higher turbidity and lower salinity due to the inflows of freshwater compared to the dry season (Figure 8). In terms of TN-TP ratio, the diatom *Cyclotella* sp. was dominant in the dry season (southwest monsoon) when phosphorus was limiting (low TN-TP ratio of 3). *Skeletonema costatum* bloomed when phosphorus concentration was high during the transition intermonsoon period, IM1 (Table 2).

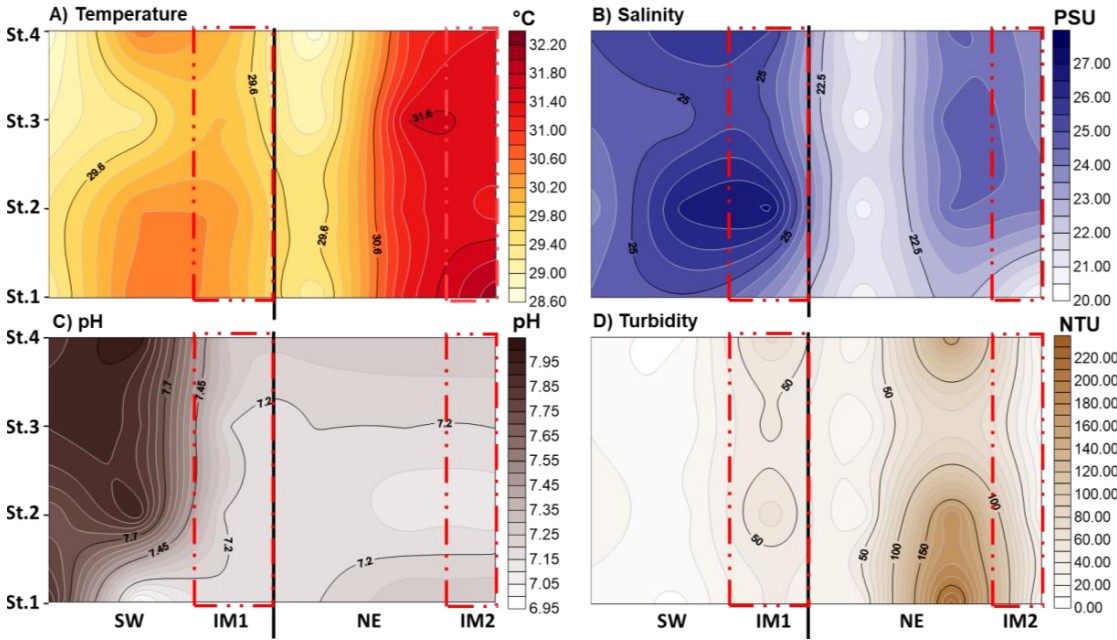

**Figure 8.** Seasonal changes of (**A**) temperature, (**B**) salinity, (**C**) pH, and (**D**) turbidity at different sites in the Sangga Kechil estuarine-mangrove area during the study period.

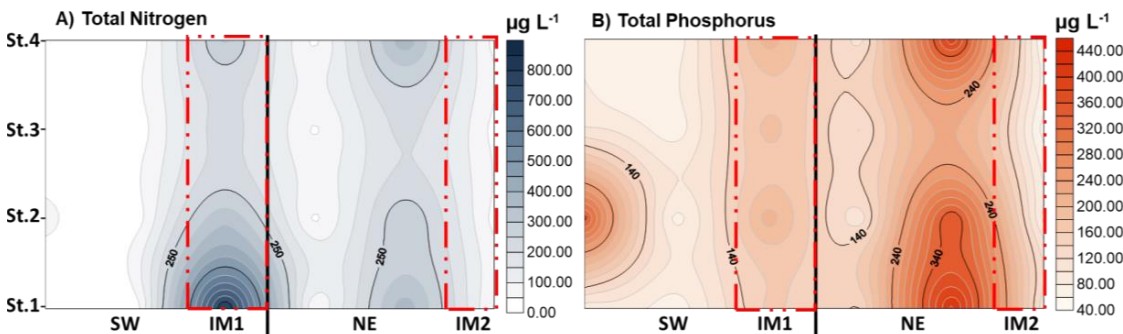

**Figure 9.** Seasonal changes of nutrients, (**A**) total nitrogen and (**B**) total phosphorus at different sites in the Sangga Kechil estuarine-mangrove area during the study period.

**Table 2.** Percentages and nutrient ratios (TN:TP) based on Redfield ratio 16:1 contrasting phytoplankton communities between monsoonal seasons in the Sangga Kechil estuarine-mangrove area.

| Phytoplankton | Southwest | Inter-Monsoon 1 | Northeast | Inter-Monsoon 2 |
|---|---|---|---|---|
| | TN:TP = 3 | TN:TP = 37 | TN:TP = 10 | TN:TP = 8 |
| Diatom (Bacillariophyta) | 92.08% | 98.90% | 66.04% | 43.68% |
| Dinoflagellates (Dinophyta) | 7.21% | 0.98% | 33.10% | 52.23% |
| Others group | 0.70% | 0.12% | 0.86% | 4.09% |
| No. of species | 64 | 34 | 54 | 53 |

Based on the principal component analysis (PCA) and CCA of eight biotic and 12 abiotic variables, main driving factors for the seasonal dynamic of major diatom species (*S. costatum*, *Coscinodiscus* sp. and *Cyclotella* sp.) were the nutrients (SRP, TAN, and TN), temperature and pH which accounted for 93.83 % of the total variance in PC1 and PC2 (Figure 10 and Table 3). The CCA biplot clearly visualized that *Skeletonema costatum* was strongly associated with the availability of nutrients, especially SRP and TAN, during the upwelling when the massive diatom blooms occurred during the post-southwest season (Figure 10). Significantly higher ($p < 0.05$) densities of *Cyclotella* sp. and *Coscinodiscus* sp. in the drier southwest monsoon were associated with pH and total ammonium nitrogen, respectively (Figure 10).

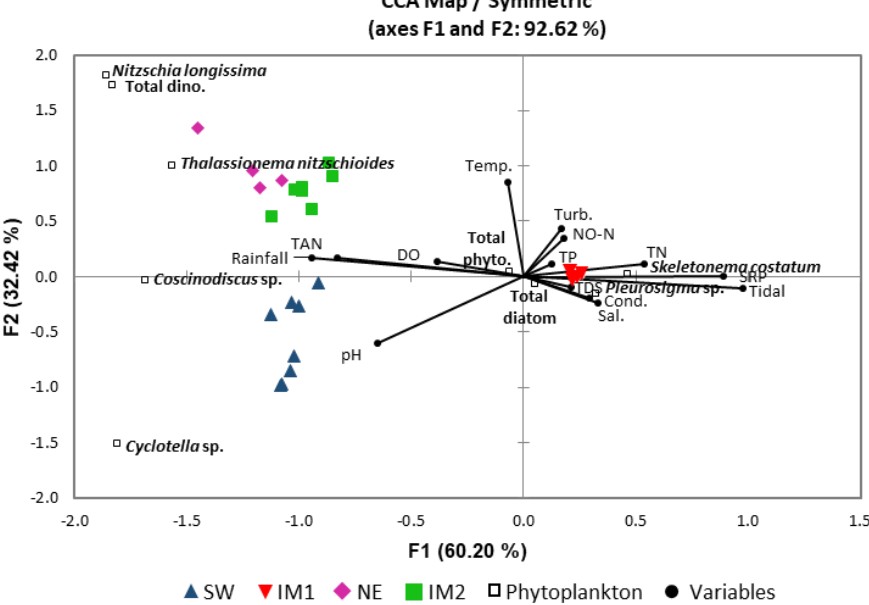

**Figure 10.** Canonical correspondence analysis (CCA) ordination of major phytoplankton groups and species with physical and chemical parameters in different seasons of the first two axes, F1 = 60.20% and F2 = 32.42%. Only the species well-characterized by the first two CCA components are shown.

**Table 3.** Inter-set correlations of significant environmental factors with the first three ordination axes of the final canonical correspondence analysis (CCA) (abundance data).

| | F1 | F2 | F3 |
|---|---|---|---|
| Eigenvalue | 0.27 | 0.14 | 0.02 |
| % of constrained inertia | 60.20 | 32.42 | 4.85 |
| % of cumulative | 60.20 | 92.62 | 97.47 |
| Total inertia | 59.19 | 31.88 | 4.77 |
| % of inertia cumulative | 59.19 | 91.07 | 93.83 |

**Table 3.** *Cont.*

|  | F1 | F2 | F3 |
|---|---|---|---|
| *Cyclotella* sp. | **0.59** | 0.41 | 0.002 |
| *Coscinodiscus* sp. | **0.83** | 0.001 | 0.04 |
| *Skeletonema costatum* | **0.99** | 0.002 | 0.01 |
| *Pleurosigma* sp. | **0.89** | 0.003 | 0.06 |
| *Nitzschia longissima* | 0.34 | 0.32 | 0.33 |
| *Thalassionema nitzschioides* | 0.40 | 0.16 | 0.00 |
| Total phytoplankton | **0.64** | 0.34 | 0.004 |
| Total diatom | 0.34 | 0.47 | 0.17 |
| Total dinoflagellates | **0.51** | 0.45 | 0.04 |
| Temperature | −0.07 | **0.85** | −0.02 |
| Dissolved oxygen | −0.39 | 0.14 | −0.16 |
| Salinity | 0.33 | −0.24 | 0.33 |
| Conductivity | 0.29 | -0.19 | 0.37 |
| pH | **−0.65** | **−0.61** | 0.009 |
| Total dissolved solids | 0.21 | -0.09 | 0.34 |
| Turbidity | 0.17 | 0.43 | 0.50 |
| Total nitrogen | **0.53** | 0.11 | 0.07 |
| Total phosphorus | 0.12 | 0.11 | 0.35 |
| Total ammonium nitrogen | **−0.83** | 0.17 | −0.07 |
| Nitrate + nitrite | 0.18 | 0.34 | −0.24 |
| Soluble reactive phosphorus | **0.89** | 0.007 | −0.14 |
| Rainfall | **−0.94** | 0.17 | −0.10 |
| Tidal level (m) | **0.98** | −0.11 | −0.02 |

PERMANOVA partitioning of seasonal phytoplankton distribution using a partial (Type III) sum of squares and pairwise comparisons showed that there were significant differences ($p < 0.05$) of total phytoplankton abundance, diatoms, and dinoflagellates among seasons except for diatoms ($p > 0.05$) during the northeast monsoon and the following transition period (Table 4).

**Table 4.** PERMANOVA results contrasting phytoplankton communities among seasons in the Sangga Kechil estuarine-mangrove area.

|  | Source | Phytoplankton | Diatoms | Dinoflagellates |
|---|---|---|---|---|
| Statistics | df | 3.20 | 3.20 | 3.20 |
|  | MS | 3487.7 | 2550.7 | 5205.7 |
|  | *F* | 6.28 | 6.31 | 6.36 |
|  | *P* | <0.001 | <0.001 | <0.001 |
| Pairwise test, P | Southwest, IM1 | 0.0022 | 0.0029 | 0.002 |
|  | Southwest, Northeast | 0.0053 | 0.0035 | 0.0003 |
|  | Southwest, IM2 | 0.0053 | 0.0033 | 0.0029 |
|  | IM1, Northeast | 0.0004 | 0.0006 | 0.0527 |
|  | IM1, IM2 | 0.0289 | 0.026 | 0.0263 |
|  | Northeast, IM2 | 0.0069 | 0.0772 | 0.028 |

df = degrees of freedom; MS = mean squares; *F* = ratio and *P* = significant levels.

## 4. Discussion

Mangrove estuaries generally support relatively high phytoplankton productivity due to naturally available nutrients from the surrounding mangrove litters that sustain the ecological balance between phytoplankton productivity and diversity [52,53]. Tanaka et al. [54] reported that nutrient outwelling from the mangrove swamp affected the phytoplankton distribution. However, nutrient inputs from anthropogenic activities into these estuaries often offset this fragile ecological balance, resulting in eutrophication and phytoplankton blooms. In this study, higher phytoplankton densities in

the mangrove estuarine waters coincided with higher nutrient availability during the upwelling, illustrating the relationship between diatoms and environmental changes, especially those related to eutrophication. Manna et al. [55], in their studies of eutrophication in Sundarban estuaries, reported that high phytoplankton density in the dry season was generally controlled by phosphate and nitrogen. Increased phytoplankton productivity during the dry season was probably due to the favourable environmental conditions related to light, temperature, and nutrient availability [56].

Diatoms and dinoflagellates are two major phytoplankton groups in marine waters, and whichever group and/or species dominates the ecosystem would depend on environmental factors that would provide the competitive advantage to the specific group. Under natural conditions, diatoms normally contribute the highest percentage compared to other phytoplankton groups in marine waters [57]. In this study, diatoms dominated the phytoplankton community (66.0% to 98.9%) during most parts of the year, but they formed only 43.7% of the total phytoplankton during the post wet season when the environmental factors favoured the dominance of dinoflagellates (Figure 6). Tanaka et al. [54] reported that diatoms dominated the phytoplankton composition in mangrove estuaries and the phytoplankton community structure was maintained by nutrients flushed from the surrounding mangroves by tidal mixing. As indicated by the CCA analysis, pH was one of the major factors influencing the phytoplankton community structure in this study. Donald et al. [58] noted that diatom frustules increased with increasing pH values. In this study, high pH in the dry season was closely associated with the dominance of *Cyclotella*, a species normally found in nutrient poor waters [59]. In addition, the dominance of diatoms in this study was also influenced by salinity fluctuations related to seasonal changes. Inputs of freshwater during the northeast monsoon into the estuary showed impacts on the salinity gradient (Figure 8), which together with the increased water turbidity subsequently caused the decline in diatom abundance (Figure 6). Khatoon et al. [60], also noted that salinity changes significantly affected the diatom growth. In addition, salinity has been reported as a major factor associated with shifts of plankton community structure in mangrove ecosystem [29,42,61]. In fact, diatoms are more predominant in estuarine areas due to high salinity and gave way to green algae in upstream when the salinity becomes low [62]. Potapova et al. [63] reported that salinity changes, which were closely related to the rainfall pattern, showed significant effects on diatom community structure. Moreover, dynamic salinity gradients in mangrove estuaries developed due to the combination of tidal currents, wind mixing, and freshwater inflow [23], which varied among monsoonal periods, particularly defined the succession of phytoplankton species.

Some species of diatoms have special environmental requirements and each species tends to become dominant when the growth conditions match its specific requirement. In our study, *S. costatum* could become an indicator for nutrient enrichments as it became dominant during the transition from the dry to wet seasons when the cooler turbulent waters caused upwelling in the estuary (Figures 5 and 7). In fact, during this time, the diatoms showed the highest mean density with *S. costatum* as the dominant species with high correlation to nutrients (Tables 1 and 3; Figure 10). Apparently, nutrient concentrations in mangrove estuaries fluctuate according to seasonal changes [64] due to variations in temperature and water column mixing. In fact, Hilaluddin et al. [29] and Lewandowska et al. [65] supported that the occurrence of the massive *S. costatum* blooms in this study was mainly affected by nutrient fluctuations that were closely related to temperature changes. According to Chen et al. [66], the composition of diatom assemblages was significantly related to shifts of temperature due to warming trends of climate change.

During the wet season, high rainfall and increased surface runoff caused an increase in the water turbidity and decrease in water transparency, resulting in reduction of light intensity and diatom density (Figures 3 and 8). Ge et al. [67] noted that suspended sediments reduced light penetration and caused a decline in diatom growth. Although the availability of nutrients concurrently influenced the diatom production, the limitation of light penetration will impede the diatom growth [68,69]. However, some species of green algae and dinoflagellates responded positively to high turbidity as they have exceptional shade adaptation [70], and some species are mixotrophic and are not affected by

light. Several dinoflagellate species have been shown to grow in various light intensities as low as 70 μmol photons m$^{-2}$ s$^{-1}$ [71].

Another potential diatom indicator was *Cyclotella* sp., which was found to dominate the phytoplankton community in the dry southwest monsoon season (Figures 5, 7 and 10; Table 3). Nutrient concentrations in the dry season were significantly lower than the wet season, but *Cyclotella* is known to require significantly lower nutrients compared to other phytoplankton species [72]. The genus *Cyclotella* has been used as a useful indicator in oligotrophic lakes [73,74], and most likely it can also be used to indicate low-nutrient conditions in coastal waters, as has been illustrated. Another diatom species that showed relatively high abundance during the dry season was *Coscinodiscus* sp. (mean density 12,668.75; 16.45%), although its presence was not as intense as that of *Cyclotella* sp. (mean density 59,106.25; 53.31%) (Figure S1). Mukherjee et al. [75] also reported that *Coscinodiscus* spp. could be a potential indicator of dissolved organic compounds in the Bay of Bengal. In addition, *Coscinodiscus* along with *Thalassionema* were reported to dominate the Sarawak estuarine waters [30]. In a Philippines mangrove estuary, a centric diatom, *Coscinodiscus wailesii,* was common during the southwest monsoon [76]. In this study, *C. wailessi* was common during the dry southwest monsoon and was almost absent during the wet season (Figure 7).

In general, diatoms were the main group of phytoplankton in the Sangga Kechil mangrove-estuary during the dry season, with a peak in the transition period (inter-monsoon) characterized by high nutrient availability due to upwelling associated with temperature change and increased water turbulence. In the wet season, high nutrient availability from autochthonous (outwelling) and allochthonous (river inflows) was masked by the high water turbidity resulting in dominance of dinoflagellates, such as *Ceratium furca* (Figure 10 and Figure S1; Table 3). With the dominance of diatoms, *Cyclotella* sp. showed the potential to be a useful indicator of the dry southwest monsoon characterized by high temperature, lower turbidity allowing higher light penetration and relatively lower nutrient. The dominance *S. costatum* succeeded the *Cyclotella* sp. when there was an increase of nutrients during the intermonsoon.

Responses of the phytoplankton community to seasonal changes in this study indicate that climate change will play a critical role in shaping phytoplankton communities in aquatic ecosystems. Alterations in monsoonal patterns, rainfall intensity, sea-level rise, ocean currents, tidal changes, waves and upwelling will affect all factors related to photosynthesis, growth, composition and diversity of phytoplankton. Predicted global change will occur gradually over decades, allowing for adaptation of phytoplankton species to become genetically and phenotypically different from the present population. Therefore, the continuity of phytoplankton community analysis should be considered to fully understand the response mechanisms associated with the association of phytoplankton succession to the phenomenon of climate change.

## 5. Conclusions

This study illustrated that the tropical phytoplankton community of Sangga Kechil estuarine-mangrove area was mainly shaped by environmental fluctuations associated with seasonal changes. Diatoms formed the most dominant group in all seasons contributing 60.0% to 80.9% of the total phytoplankton. In the late wet season, dinoflagellates succeeded the dominance as they could perform better under the high turbidity associated with high rainfall and surface runoff. Main driving factors influencing the seasonal dynamics of major diatom species (*S. costatum*, *Cyclotella* sp. and *Coscinodiscus* sp.) were the rainfall, tidal mixing, nutrients (SRP, TAN and TN), temperature and pH which accounted for 93.0% of the total variance. The diatom community structure in this estuarine-mangrove area was characterized by an indicator species associated with environmental factors driven by seasonal changes. *Cyclotella* sp. was dominant in the dry southwest monsoon characterized by high temperature, lower turbidity and relatively low nutrients. There was a shift in dominance from *Cyclotella* to *S. costatum* during the dry to wet transition period due to an increase of nutrients associated with upwelling during the intermonsoon. During the wet season, diatoms

ceased to be the major groups as the dinoflagellates have better competitive advantage in highly turbid waters. Further studies are needed to refine the selection of phytoplankton species for various environmental conditions.

**Supplementary Materials:** The following are available online at http://www.mdpi.com/2077-1312/8/7/528/s1.

**Author Contributions:** Conceptualization, F.M.Y. and F.H.; methodology, F.M.Y. and F.H.; formal analysis, F.H. and F.M.Y.; investigation F.H. and F.M.Y.; data curation, F.H.; writing—original draft preparation, F.H. and F.M.Y.; review and editing, F.M.Y., F.H., and T.T.; visualization, F.M.Y. and T.T.; supervision, F.M.Y. and T.T.; project administration, F.M.Y.; funding acquisition, F.M.Y. and T.T. All authors have read and agreed to the published version of the manuscript.

**Funding:** This research was funded by the Ministry of Higher Education Malaysia (JSPS Asian-Core Matching Grant), Core-to-Core Program (Asia-Africa Science Platforms) of the Japan Society for the Promotion of Science (JSPS) and Science and Technology Research Partnership for Sustainable Development grant (COSMOS 2016-2021), matching grant with the Ministry of Higher Education Malaysia (SATREPS COSMOS Matching Fund).

**Acknowledgments:** Special thanks to staff of the Laboratory of Marine Biotechnology, Institute of Bioscience, UPM for their technical assistance.

**Conflicts of Interest:** The authors declare that they have no known competing financial interests or personal relationships that could have appeared to influence the work reported in this paper.

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
