# Peer review of "Shifts in Diatom Dominance Associated with Seasonal Changes in an Estuarine-Mangrove Phytoplankton Community"

_jmse, doi:10.3390/jmse8070528_

Round 1
Reviewer 1 Report
This paper evaluated seasonal changes of phytoplankton community structure and environmental conditions in a mangrove estuary of Malaysia. The authors investigated phytoplankton community (abundance and composition) biomonthly during both dry (June to October) and wet (November to April) seasons at four sampling sites. This approach is pretty interesting in terms of seasonal monsoon. However, the authors failed to illustrate seasonal changes of phytoplankton community and environmental conditions as indicated the following comments.
Major comments:
- Title: should be revised in more concise way. The title, as it stands, is really naïve and is really hard to accept as an appropriate title of scientific paper.
- Sampling sites are located within the 2-km distance along the narrow corroder. Sampling frequency is not representative of climate conditions of this region. Accordingly,
- As indicated by the data points in Fig. 5, bimonthly samplings generated very restricted (i.e., 1 or 2) data points for each climate condition, Southwest monsoon, Post-SWM, Northeast monsoon, and Post-NEM. Are these 1 or 2 sampling points representative of climate conditions? That is, by one data point, how do the authors treat statistical significance between conditions? Again, by using this bimonthly sampling interval, how can the authors attest the continuity, as indicated by Figs. 7 and 8?
- Given temporal variations within the narrow ranges (as indicated by mean and SE values) of environmental variables in Table 1, really important question can be arisen “are environmental variables descriptors of changes in abundances of phytoplankton, diatoms, and diniflagellates?”
- These temporal variations within the narrow ranges finally gave birth to a separation Post-southwest monsoon data from the others in CCA (Fig. 9). For example, in F2, temperature appeared to be the most important descriptor. In reality, temperature variation is really small (1°C or less) throughout the study period (as in Fig. 7A).
Minor comments:
- References cited start by [61]. I do not follow this arranging order – alphabetically or chronologically?
- Line 23: “from of a total of 85”, delete of after from.
- Line 34, “during the year” means “throughout the year”?
- Lines 123-124, what does total ammonium nitrogen (TAN) mean? Why did the authors analyze NO3+NO2?
- Lines 168-170, the authors used MANOVA for testing statistics of biodiversity and abundance. Were those data checked for normality and homogeneity? Or, PERMANOVA should be employed.
- Lines 185-186 and Fig. 3, the authors did not mention about the methodology of the hierarchical clustering of dendrogram and data transformation, etc.
- Figures should be redrawn in more informative ways.
Author Response
Reference: JMSE-843894
Title: Shifts in Diatom Dominance Associated with Seasonal Changes in an Estuarine-Mangrove Phytoplankton Community
Responses to the Reviewers’ Comments (Use the tracked version for reference)
Suggestions/comments and authors responses -Reviewer#1
Major concern:
- This paper evaluated seasonal changes in phytoplankton community structure and environmental conditions in a mangrove estuary of Malaysia. The authors investigated the phytoplankton community (abundance and composition) biomonthly during both dry (June to October) and wet (November to April) seasons at four sampling sites. This approach is pretty interesting in terms of seasonal monsoons. However, the authors failed to illustrate seasonal changes in phytoplankton community and environmental conditions as indicated in the following comments. Title: should be revised in a more concise way. The title, as it stands, is really naive and is really hard to accept as an appropriate title of the scientific paper.
Response: Thank you for your encouraging comments. We agree with you that increased frequency would give greater details of the phytoplankton community change. But the number of samplings done over a period of one year captured most of the important events in the phytoplankton community changes.
As per your request, we have revised the title to reflect the contents.
See lines 5-7. `Shifts in Diatom Dominance Associated with Seasonal Changes in an Estuarine-Mangrove Phytoplankton Community’.
- Sampling sites: 2 km distance along the narrow corroder. Sampling frequency is not representative of climate conditions of this region.
Response: Certainly, the number of sampling frequency would provide more detailed fluctuations in the phytoplankton composition and abundance. In this study, we wanted to know the succession of diatoms over a one-year period, which essentially covers a wet and dry period. We could see the pattern of change associated with the wet and dry season, as illustrated in this study. We have redrawn the Figure 3 to highlight the monsoon seasons with the short intermonsoons.
- Figure 5: bimonthly samplings generated very restricted (i.e.,1 or 2) data points for each climate condition, Southwest monsoon, Post-SWM, Northeast monsoon and Post-NEM. Are these 1 or 2 sampling points representatives of climate conditions?
Response: In general, in this tropical region, we do not emphasize the short transition period between two major monsoonal periods, as this pattern could change from year to year. But in this study, the short period before the NE monsoon was characterized by high winds and waves, resulting in increased nutrients, probably due to upwelling in areas with enriched sediments. The sampling during this short period could catch the Skeletonema blooms, that coincided with the increased nutrients in the area.
- Again, by using this bimonthly sampling interval, how can the authors attest the continuity, as indicated by Figs. 7 and 8?
Response: Yes, we agree that the continuity here is a major assumption that we made based on our experience in this region. But you are right that under the current condition, climate change is relatively unpredictable that samplings should be done every week if resources permit. In this study, we could only carry our samplings at bimonthly intervals to see major shifts in the diatom community.
- Given temporal variations within the narrow ranges, (as indicated by mean and SE values) of environmental variables in Table 1, a really important question can be arisen “are environmental variables descriptors of changes in abundances of phytoplankton, diatoms, and dinoflagellates?”
Response: A very good question. Thank you. At this point, we can say that environmental variables have the potential to be descriptors of phytoplankton changes. Based on data obtained, there are significant relationships between the main environmental changes associated with seasons. However, further studies on a wider spatial and temporal scales are needed to confirm this contention.
- These temporal variations within the narrow ranges finally gave birth to a separation Post-southwest monsoon data from the others in CCA (Fig. 9). For example, in F2, temperature appeared to be the most important descriptor. In reality, temperature variation is really small (1ºC or less throughout the study period (as in Fig.7A).
Response: In reality, seasonal changes are directly related to the rainfall, and this sets the whole chain of events that finally characterized each season. Water temperature change is very narrow in this region. Thus, changes, however, small would probably affect the biota. In the wet season, water temperatures were in the 27 to 29 range as oppose to > 30 in the dry season – Fig. 8A.
Minor concern:
- References cited start by [61]. I do not follow this arranging order – alphabetically or chronologically?
Response: According to the format, references are arranged in alphabetical order. Thus, Shen, P.P. 2011 (61) comes later than Shen, L., 2017 (60).
- Line 23: “from of total of 85”, delete of after from.
Response: Thank you. We have corrected the mistake. See line 25.
- Line 34: “during the year” means “throughout the year”?
Response: Yes, you are right. Thank you. We have changed the phrase to “throughout the year”. See line 36.
- Lines 123-124: what does total ammonium nitrogen (TAN) mean? Why did the author analyze NO3+NO2?
Response: TAN (total ammonia nitrogen) means both unionized and ionized ammonia (NH3 and NH4+). We did not think it is necessary to separate the two forms in this study, as both forms change back and forth depending on pH and temperature. Similarly, we combine nitrate-N and nitrite-N because these forms are also dynamic changing forms from one to another. In fact, nitrite is very unstable in the environment and is readily oxidized to nitrate.
- Line 168-170, the authors used MANOVA for testing statistics of biodiversity and abundance. Were those data checked for normality or homogeneity? Or, PERMANOVA should be employed.
Response: Thank you for your comments. The data for MANOVA had been checked for normality. But we agree with your suggestion that PERMANOVA should be employed since it is more suitable for the analysis. We have updated the Table and the text in M & M accordingly. See Table 4, lines 195 -199 and lines 318-322.
- Line 185-186 and Fig. 3, the authors did not mention about the methodology of the hierarchical clustering of dendrogram and data transformation, etc.
Response: Thank you for your suggestion. We have added more information in M & M section to explain the methods. See lines 199-200.
- Figures should be redrawn in more informative ways.
Response: Most Figures had been edited and redrawn to be more informative.

Reviewer 2 Report
This study includes a thorough description of phytoplankton assemblages reconstructed by means of microscopic enumeration. The authors examined key potential drivers of phytoplankton community structure and concluded that environmental variables “water temperature, nutrients, pH” accounted for most of the variance in the system.
The work appears to have been carefully conducted and data thoroughly analyzed. As such the data were clearly presented but the results and discussion section can improve from some additional discussions particularly in regard to the multivariate analysis.
One interesting aspect of this work is that is it showed the importance of species succession under different environmental conditions, the manuscript can improve if the authors show relationship with the change in nutrient ratios and phytoplankton species and groups. Tidal fluctuation in an estuarine environment is an important parameter and there no mention of on the role of tide in the estuary. Authors also mention upwelling several times in the manuscript; it was not clear which or what parameter was used to determine upwelling. The data shown in the figures does not clearly show any upwelling event. Same goes with turbulence. The results also suggest that pH was an important parameter however, there is little to no discussion of how pH can affect phytoplankton community and additionally why we are observing the change in pH in the estuary and its role in community dynamics? Some additional comments on how future climate changes can play a role in shaping phytoplankton community in mangrove systems.
Overall, the authors did a nice work and there is useful information provided in this manuscript. However, the results are largely confirmative of studies from other mangrove estuarine systems. I recommend revision following the suggestions mention above.
Author Response
Reference: JMSE-843894
Title: Shifts in Diatom Dominance Associated with Seasonal Changes in an Estuarine-Mangrove Phytoplankton Community
Responses to the Reviewers’ Comments (Use the tracked version for reference)
Suggestions/comments and authors responses -Reviewer#2
Major concern:
- As such the data were clearly presented but the results and discussion section can improve from some additional discussions particularly in regard to the multivariate analysis.
Response: Thank you for your comments. We have added more information in the Result and Discussion section with regards to the multivariate analysis. See Table 4, lines 195-199, and lines 318-322.
2. One interesting aspect of this work is that is it showed the importance of species succession under different environmental conditions, the manuscript can improve if the authors show a relationship with the change in nutrient ratios and phytoplankton species and groups.
Response: We agree with your suggestion. We have added more information to show the relationship with the change in nutrient ratios, and concomitantly the phytoplankton groups and species. See lines 295-298 and Table 2.
- Tidal fluctuation in an estuarine environment is an important parameter and there no mention of on the role of tide in the estuary.
Response: We agree with your suggestion on the importance of tidal fluctuations in an estuarine environment. We have added general information on tidal currents in the introduction section, and also provided tidal levels and water currents in the result section. See Figure 2, lines 67-79 and lines 134-138, Table 1, and Table 3.
- Authors also mention upwelling several times in the manuscript; it was not clear which or what parameter was used to determine upwelling. The data shown in the figures does not clearly show any upwelling event. Same goes with turbulence.
Response: Upwelling was related to the increase in nutrients, especially TN in the water column, most probably from the enriched bottom water (Fig. 9).
- The results also suggest that pH was an important parameter, however, there is little to no discussion of how pH can affect phytoplankton community and additionally why we are observing the change in pH in the estuary and its role in community dynamics?
Response: That is true. Thanks for pointing out this shortcoming. pH, like salinity, decreased during the wet season due to the inflow of freshwater into the estuary. pH was high in the dry months and was related to Cyclotella. We have revised the relation text accordingly and added more information on the role of pH in the phytoplankton community dynamic.
See lines 371-375.
6. Some additional comments on how future climate changes can play a role in shaping phytoplankton community in mangrove systems.
Response: We agree with your suggestion. We have added some information on the potential impacts of climate change to the phytoplankton community. See line 434-448.

Round 2
Reviewer 1 Report
The original version of the manuscript has been deeply revised greatly increasing its clarity and general quality; under the present conditions is acceptable for publication in JMSE.
Reviewer 2 Report
I have no further comments. Satisfied with the revised manuscript.